# Fine Carbohydrate Structure of Dietary Resistant Glucans Governs the Structure and Function of Human Gut Microbiota

**DOI:** 10.3390/nu13092924

**Published:** 2021-08-24

**Authors:** Arianna D. Romero Marcia, Tianming Yao, Ming-Hsu Chen, Renee E. Oles, Stephen R. Lindemann

**Affiliations:** 1Whistler Center for Carbohydrate Research, Department of Food Science, Purdue University, West Lafayette, IN 47907, USA; aromerom@purdue.edu (A.D.R.M.); yao132@purdue.edu (T.Y.); minghsu.chen@ntu.edu.sg (M.-H.C.); roles@purdue.edu (R.E.O.); 2Department of Nutrition Science, Purdue University, West Lafayette, IN 47907, USA

**Keywords:** resistant glucans, dietary fiber, polysaccharides, polydextrose, SCFAs

## Abstract

Increased dietary fiber consumption has been shown to increase human gut microbial diversity, but the mechanisms driving this effect remain unclear. One possible explanation is that microbes are able to divide metabolic labor in consumption of complex carbohydrates, which are composed of diverse glycosidic linkages that require specific cognate enzymes for degradation. However, as naturally derived fibers vary in both sugar composition and linkage structure, it is challenging to separate out the impact of each of these variables. We hypothesized that fine differences in carbohydrate linkage structure would govern microbial community structure and function independently of variation in glycosyl residue composition. To test this hypothesis, we fermented commercially available soluble resistant glucans, which are uniformly composed of glucose linked in different structural arrangements, in vitro with fecal inocula from each of three individuals. We measured metabolic outputs (pH, gas, and short-chain fatty acid production) and community structure via 16S rRNA amplicon sequencing. We determined that community metabolic outputs from identical glucans were highly individual, emerging from divergent initial microbiome structures. However, specific operational taxonomic units (OTUs) responded similarly in growth responses across individuals’ microbiota, though in context-dependent ways; these data suggested that certain taxa were more efficient in competing for some structures than others. Together, these data support the hypothesis that variation in linkage structure, independent of sugar composition, governs compositional and functional responses of microbiota.

## 1. Introduction

Although diet is increasingly understood to play a major role in modulating the gut microbiome [1,2], the mechanisms by which this occurs are still unclear. Human diets vary widely across populations [3,4], and even within a single individual over relatively short periods. Thus, the diversity and idiosyncrasy of diets makes it challenging to identify how the individual components thereof influence gut microbiome structure and function in predictable ways across individuals. One class of food components known to significantly shape the gut microbiome’s structure and function, and, thereby, human health, is fermentable dietary fibers, which are resistant to hydrolysis by human enzymes but are degraded by colonic microbes, increasing and maintaining diversity of the gut microbiome [5]. The task of linking fiber polysaccharides with predictable gut microbiome responses is made more difficult in that complex carbohydrates vary in both composition (i.e., the types and ratios of sugars that compose the polymer) and in structure (i.e., how those sugars are linked via glycosidic bonds). Specifically, polysaccharides can vary across multiple structural dimensions, including monosaccharide composition, anomeric configurations, glycosidic linkages, linear chain lengths, and branch chain compositions [6]. This diversity generates a wide range of possible higher-order polymer structures [7]; for example, a pentasaccharide is estimated to have over 1.5 billion possible structural forms [8]. Because glycans are so heterogeneous and these structural properties (i.e., composition and structure) covary in most naturally derived fiber polysaccharides, it is very challenging to separate the effects of composition and structure on gut microbiota using fibers extracted directly from plants.

The gut microbiome has recently been shown to be very sensitive to even subtle structural differences in both insoluble [9] and soluble [10] plant-derived fibers; this has supported the hypothesis that discrete fiber structures may target certain microbial taxa with wide variation in specificity [11]. Because the enzymes required for degradation of complex carbohydrates are highly specific to their cognate glycosidic linkages, and possibly to higher-order structures [12,13], there exists the potential for organisms to specialize in hydrolyzing specific bonds and/or consuming specific parts of the molecule [14]. If true, this niche partitioning may allow cooperative consumption of polysaccharides, in which organisms avoid competition through division of metabolic labor [15] and, thereby, increase the microbial diversity that can be sustained.

Here, we aimed to test the related hypotheses that (1) complex carbohydrates identical in composition (i.e., composed solely of glucose) but varying in structure (i.e., with different structural parameters) would select for distinct microbiota; and (2) that the same microbes would be selected by identical structures across different individuals’ microbiota. To test this hypothesis, we used commercially available glucans, generated either by modification of starches (mixed linkage α-glucans and maltodextrins) or polymerization of glucose (polydextroses) by heating or enzymatic catalysis to be at least partially resistant to degradation by human enzymes (together, here collectively termed “resistant glucans”). Resistant glucans are made by multiple companies with varying sugar and starch sources and processes, and are commonly added to food products in order to increase dietary fiber content to improve health outcomes [16,17,18,19,20]. We performed in vitro fermentation of 11 distinct resistant glucan products using fecal microbiota from three fecal donors (individually) and measured community structure and metabolic function (pH, gas, and short-chain fatty acid (SCFA) production) over time. Our results suggest that, although different initial microbiota respond divergently at the whole-community level with respect to metabolism and overall community structure, specific taxa respond strongly to certain glucan structures across all individuals’ microbiota. These data suggest that the fine structure of a carbohydrate alone, independent of differences in its sugar composition, can target certain microbial taxa and alter community structure in similar ways across individuals.

## 2. Materials and Methods

### 2.1. Fibers Used

The resistant glucans and one fructan used in this study were gifts from several companies (Tate & Lyle, Hoffman Estates, IL, USA; Ingredion, Westchester, IL, USA; Archer Daniels Midland, Chicago, IL, USA; Samyang, Seoul, Korea). These glucans ranged in moisture content from 2–8% (dry weight), which was measured by a Mettler Toledo Moisture Analyzer HE53 (115V) (Columbus, OH, USA). Masses added for all analytical procedures and microbial cultures were adjusted for moisture content to ensure equivalent carbohydrate loading.

### 2.2. Carbohydrate Linkage Analysis

The glycosidic linkage composition of the glucans was determined via the partially methylated alditol acetate approach and measured using gas chromatography (model 7890A, Agilent Technologies Inc., Santa Clara, CA, USA) with a fused capillary column (SP-2330, Supelco Analytical, Bellefonte, PA, USA) coupled with a mass spectrometer (model 5975C, Agilent Technologies Inc., Santa Clara, CA, USA) as previously described by Pettolino et al. [21] and Xu et al. [22] and analyzed as in Tuncil et al. [23]. Briefly, reactions started with 0.5–1 mg of sample and were methylated with CH_3_I and hydrolyzed with 2M TFA at 121 °C. Myo-inositol (20 μL of 2.5 mg per mL 2M TFA) was added as an internal standard. Prior to the reduction of the samples with acetic acid, they were treated with 2M NH_4_OH and 1M freshly prepared NaBD_4_ in 2M NH_4_OH. Acetylation was performed with the addition of acetic anhydrate, followed by dichloromethane and multiple ddH_2_O washes, and the residue was dissolved in acetone. The GC-MS conditions were as follows: injector volume, 1 μL; injector temperature, 240 °C; detector temperature, 300 °C; carrier gas, helium: 1.9 m/second; split ratio, 100:1 and the temperature program, 100 °C for 2 min, 8 °C/min to 240 °C for 20 min.

### 2.3. Fecal Sample Collection

Feces from three healthy donors were collected for inocula. Donors were selected from healthy adults 18–65 years of age, of normal or overweight BMI (18.5 kg/m^2^ < BMI < 30 kg/m^2^) who had not experienced significant dietary change in the past two weeks and who resided within 30 miles of Purdue University. Donors that were pregnant or lactating, had a history of gastrointestinal or chronic metabolic disease, had had major surgery of the gastrointestinal tract, or had taken prebiotic or probiotic supplements in the prior three months or antibiotics in the prior six months were excluded. All selected donors were male with an omnivorous diet and reported distinct dietary patterns. Donor 1 was a 25-year-old with a BMI of 31.7 and reported frequent consumption of high-fat and spicy foods. Donor 2 was a 22-year-old with a BMI of 24.9 and reported a diet with elevated consumption of fruits and vegetables. Donor 3 was a 22-year-old male with a BMI of 25.4 and reported a diet with high fermented dairy consumption. Involvement of human subjects in this study was reviewed and approved by Purdue University under the IRB Protocol #: 1701018645.

### 2.4. In Vitro Fecal Fermentation

For each donor’s microbiota, collected samples were immediately placed into ice post-collection and rapidly transferred to an anaerobic chamber (Coy Laboratory Products, Grass Lake, MI, USA), where they were processed as previously described [15]. The basal media (phosphate buffer) contained NaCl, KCl, urea as the sole nitrogen source, Na_2_SO_4_, resazurin, Na_2_HPO_4,_ and were autoclaved together; heat sensitive compounds such as CaCl_2_, MgCl_2_, 1000X P1 metal solution and cysteine hydrochloride were added post-cooling. Once inside the chamber, fecal samples were mixed with an anoxic 40 mM sodium phosphate buffer with pH 6.95 in a 1:20 ratio (*w*/*v*) and filtrated through four layers of cheese cloth to make a slurry. Glucans were dissolved in sterile water, passed through a 0.22 μm Millipore syringe filter and mixed 1:1 with anoxic 2X phosphate buffer to a concentration of 1.25% *w*/*v*. This solution was then mixed with the fecal slurry to a 4:1 ratio into Balch tubes for each time point in triplicate, with a total carbohydrate concentration of 1% *w*/*v*. Tubes were closed with butyl rubber stoppers and aluminum seals and incubated outside the chamber in an incubator at 37 °C shaking at 150 rpm. Culture tubes were taken out of the incubator at each time point (4, 8, 12, 24, 36, and 48 h) and pH and gas measurements were taken as previously described [24]. Gas was measured by overpressure with an 18 ga needle attached to a 10 mL glass syringe; tubes were unsealed and two aliquots were taken out and stored at −80 °C for further SCFA analysis and DNA extraction. pH was measured using a Mettler Toledo pH meter.

### 2.5. Short-Chain Fatty Acid (SCFA) Analysis

Short-chain fatty acids were analyzed by gas chromatography with a flame ionization detector (GC-FID 7890 A Agilent Technologies Inc., Santa Clara, CA, USA) with a fused silica capillary column (Nukon SUPELCO, Bellefonte, PA, USA) as previously described [24] and under the following conditions: front detector temperature at 230 °C, oven temperature at 100 °C and helium as carrier gas. Frozen samples were thawed and centrifuged for 10 min at 13,000× *g* rpm. 400 μL of supernatants were mixed with 100 μL of an internal standard (4-methylvaleric acid in 6% *v*/*v* phosphoric acid and copper sulfate pentahydrate). The external standards used were acetic, propionic, butyric, isobutyric, and isovaleric acids mixed in equimolar proportions.

### 2.6. DNA Extraction

DNA was extracted using the FastDNA SPIN extraction kit (MP Biomedicals, Santa Ana, CA, USA). The aliquots taken at 48 h were thawed, centrifuged, and the pellet was re-suspended. The process was followed as the FastPrep 24 protocol indicates with the following modifications: bead beating time (twice for 40 s at 6.0 m/s) and triplicating shaking time of the sample and binding matrix (shake 15 min instead of 5 min). The extracted genomic DNA was stored at −20 °C until further sequencing analysis.

### 2.7. 16S rRNA Sequencing

DNA was quantified and normalized to 100 ng/μL. The V4–V5 region of the 16S rRNA gene was amplified by PCR using the universal bacterial primers: 515-FB forward (GTGYCAGCMGCCGCGGTAA) and 926-R reverse (CCGYCAATTYMTTTRAGTTT) [25] and KAPA HiFi Hot Start ReadyMix

The following thermal cycler conditions: initial denaturation, 95 °C for 5 min, 20 cycles of denaturation (98 °C, 20 s), annealing (60 °C for 15 s), and extension (72 °C for 30 s), and a final extension (72 °C, 10 min) followed by an infinite hold at 4 °C. PCR products were purified and prepared as previously described by Tuncil et al. [24]; briefly, after the first amplification, unincorporated primers and nucleotides were removed using 1.2X the PCR volume of the AxyPrep Mag PCR Clean-up beads according to the protocol, then quantitated and normalized with TruSeq indexed primers (IDT, Coralville, IA, USA) followed by a second cleanup using a bead volume of 1.8X the PCR volume. Post-cleanup DNA was quantified using the Qubit dsDNA HS Assay Kit (Invitrogen, Carlsbad, CA, USA) according to the manufacturer’s protocol. DNA with similar concentrations were then pooled into 11 pools for sequencing. Pools were examined for size using a BioAnalyzer (Agilent, Santa Clara, CA, USA) and quantified using the KAPA Library Quantification Kit (Roche Diagnostics, Indianapolis, IN, USA) prior to sequencing on a 2 × 250 cycle Illumina MiSeq run at the Purdue Genomics Core Facility. Sequence data are available in the National Center for Biotechnology Information’s Sequence Read Archive under BioProject PRJNA721693 as BioSamples SAMN18744139–SAMN18744255.

### 2.8. Sequence Processing and Community Analyses

Sequences were processed with mothur (v.1.39.3) [26] following the MiSeq SOP (https://mothur.org/wiki/miseq_sop/ (accessed on 24 August 2021)) with the following modifications. Sequences were screened for a maximum length of 411 nt, zero maximum ambiguous bases and a maximum homopolymer length of 9 nt, then aligned to the mothur-formatted SILVA reference alignment across positions 13,862 to 27,654. Sequences were classified based on the Ribosomal Database Project reference training set with an 80% cutoff. Chloroplast, Mitochondria, Archaea, Eukaryota and unknown classifications were removed from further processing. OTU classifications at the species levels are reported as percentage of reads based on a 97% similarity. Ecological metrics were calculated within mothur: α-diversity metrics were calculated using the simpsoneven, chao, invsimpson and sobs calculators and β-diversity metrics were calculated using the jaccard and thetayc calculators as implemented in mothur. Distance matrices for β-diversity metrics based on the pcoa command in mothur were plotted for visualization using R. Abundance changes were log_2_-transformed for the most-abundant 33 OTUs.

### 2.9. Statistical Analyses

All analyses were performed in triplicate. SCFA data are presented as means and standard deviation. Tukey HSD at α = 0.05 was used to compare mean differences. Analysis of molecular variance (AMOVA) tests command in mothur were also computed between glucan classes to determine whether centroids were significantly distinct. Alpha-diversity was determined by One-way ANOVA *p* < 0.0001 with a Kruskal-Wallis test for multiple comparisons. Linear discriminant effect size (LEfSe) analysis was conducted using LEfSe v. 3.12 [27]. OTUs correlation relationships were determined with nonparametric Spearman using GraphPad Prism 8.4.2.

## 3. Results

### 3.1. Resistant Glucans Produced by Different Methods Varied Significantly in Structure

Glucans provided for this study came from different commercial sources and were categorized into three different groups based upon their source and structural characteristics. Significant differences in glucan structure were revealed by carbohydrate linkage analysis (Figure 1). These glucans were categorized as mixed linkage α-glucans (A–C), resistant maltodextrins (D, G and J–M), and polydextroses (E and H). These glucans are completely or semi-synthetic, non-digestible carbohydrates that meet the FDA dietary fiber definition [28]. Resistant maltodextrins and mixed linkage α-glucans are generated from starch hydrolysis products, and hence are composed of human-digestible linkages (α−1,4, α−1,6) as well as bonds that cannot by hydrolyzed by human enzymes, such as α−1,2, α−1,3, and some beta linkages. In contrast, polydextrose is polymerized from monomeric glucose and is composed of random α− and β−1,2, −1,3, −1,4 and −1,6 linkages. Interestingly, even within categories, glucans differed in the type and amount of multiply branched glycosyl residues and spanned a range of chemical complexity (defined by the number of distinct carbohydrate linkages). Of the set, the mixed linkage α-glucans A–C can be classified as relatively simpler polymers (with less modification of the native starch structure and lacking evidence of multiply linked branches), compared to glucans E, H, D, G and J–M. These more complex glucans revealed a higher diversity of carbohydrate linkages, both with respect to distinct linkage types and abundances of multiply branched glycosyl residues. The three mixed linkage α-glucans spanned a complexity gradient, with glucan A the most complex and C resembling, in linkage profile, native starch structure. However, it is important to note that anomeric configuration of these linkages is not indicated via the partially methylated alditol acetate method; thus, a fraction of the 4- and 6-linked glucose in these glucans may occur in beta linkages, which are also indigestible by human enzymes.

### 3.2. Community Metabolic Outputs from Identical Glucans Diverges across Distinct Microbiota Compositions

We tested multiple glucans with different configurations to determine how similar or different metabolic outputs would be in fermentation by fecal microbiota from each of three different healthy donors. Throughout in vitro batch fermentation, we measured pH, gas production, and concentrations of the most common short-chain fatty acids (SCFAs; acetate, butyrate and propionate) at intervals over a 48 h time course.

As expected, we observed significant gas and acid production from all glucans, although the magnitude varied across glucan classes and individual donor microbiota (Figure 2). Although all glucans were strongly fermented, terminating at pHs near or below 5, the mixed linkage α-glucans group fermented more rapidly (as evidenced by a more rapid pH drop and more rapid evolution of gas), especially glucan C. Whether these glucans fermented strongly to acid or gas was donor-dependent and, as expected, acid and gas production were generally inversely related. The decrease in pH was especially marked in fermentation of glucans A–C by donor 3 microbiota; conversely, donor 2 microbiota produced significantly more gas from the same mixed linkage α-glucans, excepting donor 1 microbiota’s strong response to glucan A. In contrast, fermentations of polydextroses and resistant maltodextrins decreased in pH much less rapidly for all donors. Interestingly, gas and acid production rates were microbiota- and glucan-specific, suggesting initial microbiome structures were differently poised for specific glucans and their corresponding metabolic outputs.

Despite similarity in overall gas and acid production for most glucans, patterns of SCFA production from different structures varied substantially across donors’ microbiota (Figure 3). Although same-glucan patterns of acetate production were similar across all the donors’ microbiota, the magnitude of acetogenesis varied substantially across individuals; however, within-donor acetate production from the mixed linkage α-glucans was stronger than for any other glucan. In contrast, patterns of propiogenesis and butyrogenesis were glucan- and donor-specific. Donor 1 microbiota were generally the most butyrogenic on all substrates except the mixed linkage α-glucans; on these substrates, donor 1 microbiota were more strongly propiogenic than butyrogenic. Conversely, donor 2 microbiota were by far the most propiogenic on polydextroses and resistant maltodextrins but were very strongly butyrogenic and less propiogenic in consumption of mixed linkage α-glucans. Within glucan categories, SCFA production from distinct glucans varied, especially for donor 3 microbiota. Generally, the SCFA output of donor 3 microbiota was lower than either of the other two, especially on mixed linkage α-glucans and for acetate and propionate; however, on some substrates (i.e., polydextrose E and resistant maltodextrin G) butyrate production from donor 3 microbiota was equivalent to those of other donors and substantially lower on others (i.e., resistant maltodextrin J).

Despite these idiosyncrasies, we observed some general tendencies in fermentation of different glucan classes. Across donors, the resistant maltodextrin category produced the most butyrate, whereas the mixed linkage α-glucans produced the least butyrate or propionate but the most acetate (*p* < 0.05). Together, these data suggested that, although distinct glucan structures were differently metabolized by different microbiota, which produced different metabolic outputs at equivalent total carbohydrate loadings, the eventual fate of glucan metabolism emerged largely from initial microbial community structure.

### 3.3. Glucan Structure Affects the Community Structure, but Not Diversity, of Fermenting Microbiota

We assessed changes in microbial abundances after 48 h of fermentation using 16S rRNA gene amplicon sequencing. Differences in α-diversity among glucan fermentations were strongly driven by initial differences in donors’ fecal microbiota. With Good’s coverage above 97% for all donors’ fecal samples, donors varied in overall α-diversity and both their richness (species observed) and evenness (Simpson’s evenness) components. Most notably, donor 1 microbiota displayed significantly lower overall diversity compared with the other two donors (*p* < 0.0001) as determined using the inverse Simpson index, and donor 2 and donor 3 microbiota were not significantly different (*p* = 0.09). However, donor 2’s initial community was significantly higher in richness compared to the other donors (*p* < 0.0001). Donor 1’s initial community was significantly less even when compared with the other two donors (*p* < 0.0001), however donor 2 and donor 3 did not significantly differ in evenness (Figure 4). By comparison, differences among glucans in post-fermentation α-diversity were minor. From the fecal inoculum, decreases in the inverse Simpson index and its richness and evenness components were observed for fructan controls and mixed linkage α-glucan cultures with donor 1 microbiota and for resistant maltodextrin D, G, and L cultures with donor 2 microbiota (*p* < 0.05) (Figure 4). No significant differences were observed among glucans or with respect to the fecal inoculum for donor 3 microbiota.

Although all initial donor microbiomes were dominated by members of phylum *Firmicutes* (53%, 63%, and 54% for donors 1,2, and 3, respectively), they displayed very different initial microbiota structures. Each had at least one dominant OTU from phyla other than *Firmicutes*; OTU00001 (*Prevotella copri*, 37%) and OTU00006 (*Faecalibacterium prausnitzii*, 20%) dominated the donor 1 community, OTU00003 and OTU00004 (both *Bacteroides* spp.) constituted ~26% of the reads from the donor 2 community, and OTU00002 (*Bifidobacterium* spp., 25%) and OTU00006 (*F. prausnitzii*, 16%) were dominant in the donor 3 community. These initial community structures governed relative abundances in fermentations (and, therefore, clustering based upon β-diversity) being much more influential than glucan structure (Figure 5). Consequently, although centroids of clusters based upon glucan class were not significantly different across all donors, significant differences were observed within donors (AMOVA, *p* < 0.001; Table 1).

However, some similarities in overall community composition responses were observed across donors. When considered at the level of the individual donors, clusters separated based upon glucan category for each. Interestingly, as observed with respect to metabolic outputs, community structures of microbiota fermenting resistant maltodextrins and polydextroses were more similar to one another than those fermenting mixed linkage α-glucans (which were, in turn, more similar to fructan and glucose controls).

### 3.4. Glucan Structural Variants Selected for Specific OTUs in a Context-Dependent Manner

Although changes in overall community structure were observed, these changes were relatively minor over the 48 h fermentation. Relative abundances of OTUs in post-fermentation communities strongly resembled the initial community structures, structuring significantly more strongly by donor than by glucan (Figure 6). However, some broad patterns of OTU responses to glucans across donors could be discerned. Notably, mixed linkage α-glucans supported large populations of OTU00002 (*Bifidobacterium* spp.) across donors, although the population sizes were linked with α-glucan complexity (increasing in abundance with decreasing complexity) only with donor 2 microbiota. Additionally, members of order *Bacteroidales* (here, genera *Bacteroides, Prevotella,* and *Parabacteroides*) and family *Lachnospiraceae* (here, genera *Anaerostipes, Roseburia, Fusicatenibacter,* and *Blautia*) were very abundant and diverse in fermentations of polydextroses and/or resistant maltodextrins; however, the most abundant OTUs within these group varied by donor, largely based upon initial population sizes. Members of family *Ruminococcaceae* generally decreased in abundance from the inoculum irrespective of substrate. In general, however, community compositional responses to glucans appeared context-dependent and largely idiosyncratic to donors.

Despite the context-dependent responses in community structure, linear discriminant analysis revealed significant associations of taxa with distinct glucan structures across donors’ initial community structures (Figure 7). When glucans were considered together in their classes, members of *Bacteriodaceae* were linear discriminants of resistant maltodextrins, whereas members of families *Porphyromonadaceae*, *Erysipelotrichaceae*, *Veillonellaceae*, and *Lachnospiraceae* were discriminants of polydextroses (LDA ≥ 3.5). However, within families *Bacteroidaceae* and *Lachnospiraceae*, many genera and species exhibited preferences for the other classes of glucans; for example, *Bacteroides ovatus* and *Bacteroides uniformis* were discriminants of polydextroses, whereas *Fusicatenibacter saccharivorans* and *Roseburia faecis* were discriminants of resistant maltodextrins (Appendix A). We also observed significant preferences of different taxa for distinct glucans within categories; for example, although most members of genus *Bacteroides* (including *B. ovatus* and *B. uniformis*) were most abundant in fermentations of resistant maltodextrin G, *B. faecichinchillae* was overrepresented in fermentations of resistant maltodextrin M (Appendix A). Similarly, resistant maltodextrin D favored *Fusicatenibacter saccharivorans*. The two polydextroses also favored different taxa; polydextrose E was preferential to *Parabacteriodes distasonis* and *Blautia wexlerae*, and *Anaerostipes hadrus* and *Clostridium ramosum* were discriminants of polydextrose H. Together, these data suggested that organisms responded distinctly to resistant glucans with subtle variations in structure. Interestingly, we found no discriminants of the mixed linkage α-glucans; this result may be explained by less specific consumption of these glucans by certain OTUs [11] and, therefore, less significant community structure variations from the initial inoculum or positive controls.

To determine whether OTU-level specificities for distinct glucans were present but obscured by idiosyncratic donor responses, we compared OTUs by their change in abundance over fermentation in all three donor contexts. In this analysis, organisms growing faster than average display increases in relative abundance, while those growing slower than average will be seen to decrease in abundance. This revealed both similarity and context-dependence in OTU responses to different glucan structures and, further, differences in growth patterns in different OTUs classified within the same genus (Figure 8). For some OTUs, such as OTU00016 (*Parabacteroides*), responses were glucan-specific and similar across all donors (for this OTU, revealing most notably consistently stronger responses to polydextrose E (increasing 3.7, 3.5, and 20-fold for donors 1, 2, and 3, respectively) than H, but similar preferences for the different resistant maltodextrins). However, for other species, responses to distinct glucans were strongly influenced by donor of origin. For example, although OTUs within genus *Bacteroides* responded with generally above-average growth on polydextroses and resistant maltodextrins across donors, they responded to mixed linkage α-glucans with above average growth rates uniformly in the context of donor 3’s microbiota and solely to glucan A in donor 1 and donor 2 communities. Further, different OTUs within *Bacteroides* displayed very different growth patterns on the resistant maltodextrins and polydextroses; for example, in donor 1 and 3 microbiota, OTU00003 responded more strongly to polydextroses than resistant maltodextrins, but vice versa in donor 2 microbiota. OTU00008 showed a similar pattern to OTU00003 for donors 1 and 2, but in donor 3 it was similarly responsive to all polydextroses and resistant maltodextrins. Similar OTU-level differences in glucan response based upon community context were observed within genus *Blautia* for all donors. Additionally, analogous dynamics occurred where a single dominant OTU was identified in a genus; OTU00005 (*Fusicatenibacter* sp.) responded strongly to polydextrose H but not to polydextrose E or resistant maltodextrins in donor 1 communities, but grew at an above-average rate across all glucans in context of donors 2 and 3. OTU00010 (*Anaerostipes* sp.) responded strongly to polydextrose H and resistant maltodextrins J-M in donor 1 communities, only to polydextroses (especially H) in donor 2 communities, and did not record above-average growth on any glucan in donor 3 microbiota. OTU00027, a *Clostridium* XIII sp., showed above-average growth on mixed linkage α-glucans in a donor 2 community and below-average growth in the community of donor 1; however, this organism grew at above-average rates on polydextroses and resistant maltodextrins (except for D, G, and J in the context of donor 2 microbiota only). Taken together, these data suggest that organismal preferences for distinct glucan structures exist, but that actual species’ responses depend upon (1) strain-level differences among donors of origin that cannot be resolved in OTUs or (2) interaction with other community members (both positive and negative).

Cross-donor abundance fold change correlations revealed glucan-specific patterns. In many cases OTUs (and, sometimes, higher taxa) behaved similarly within (or even across) glucan classes (Appendix A). For example, members of *Bacteroides* (OTU00017, OTU00003, OTU00004, OTU00008, OTU000014, and OTU00011) were generally strongly correlated with one another across glucan classes, with the exception of moderate inverse correlation in growth between OTU00003 and both OTU00017 and OTU00004 on polydextrose H and resistant maltodextrins J and M (Figure 9). However, for some OTUs (and higher taxa), the correlations in abundance fold change among members were strongly determined by substrate glucan. Similarly, in genus *Blautia* (OTU00033, OTU00007, OTU00026, OTU00023, OTU00019, and OTU00062), the level of coherence in growth response on glucans among the various OTUs depended upon the community of origin (Figure 9). In donor 1 communities, OTU00026 displayed significantly above-average growth (compared with other members of *Blautia*) on essentially all polydextrose and resistant maltodextrin substrates. This OTU also showed above-average increases in relative abundance in donor 2 communities. However, OTU00007 showed above-average growth on resistant maltodextrins K, L, and M and OTU00062 responded strongly to poly-dextroses E and H and resistant maltodextrins D, G, and J (though still composing a small fraction of the community, owing to its small initial abundance). On the other glucans, the growth responses of these two OTUs were average or below. In donor 3 communities, only OTU00007 and OTU00026 show slightly faster growth than their other cousins in *Blautia*, though this effect was much smaller across glucans than in donor 1 and 2 communities. Taken together, these data suggest that interactions among OTUs and complex glucans depended upon (1) the fine structure of the glucan; and (2) the initial abundances of other community members.

## 4. Discussion

In nature, variation in polysaccharide structure is overwhelmingly associated with differences in the ratios of glycosyl residues that compose the polymer (starch and glycogen being notable exceptions) [29]. For example, arabinoxylans extracted from bran of three classes of wheat exhibited coordinate variation in sugar composition and linkage structure [10]. These variables may exert independent influences upon microbial community responses to complex substrates. Therefore, in this study we aimed to separate these two variables by using glucans that varied in fine structure but were composed entirely of glucose, as substrates for gut microbiota in experimental fermentations. Although some natural homoglycans (for example, amylose and amylopectin) contain just one or two linkage types and have a relatively simple chemical structure (even sometimes being large polysaccharides), the glucans tested here displayed significant variability in linkage complexity despite being composed of a single sugar. In principle, different linkage structures among glucans may distinctly impact microbiome structure and function due to enzyme specificities which vary among organisms and thereby target different microbes [11]. Differences in glycosidic linkage types even among glucose disaccharides has been demonstrated to significantly alter the SCFA outputs of fecal microbiota [30], suggesting that even these simple glucans are metabolized by different microbes. Furthermore, in consumption of disaccharides, differences in transport are unlikely to be very influential; in contrast, as glucan sizes increase, the potential for specialization around transport of specific types of oligosaccharides, in addition to hydrolysis of specific linkages, and to govern division of labor and maintenance of diversity in microbiomes increases [31]. Broadly, our data supported the hypothesis that variation in fine polysaccharide structure, independent of variation in the sugars that compose the polymer, results in different composition and function of fermenting gut microbiota.

Our results demonstrate that microbiome responses to structurally distinct resistant glucans depend upon both fine glucan structure and community context, and community metabolic phenotypes emerge from the interaction of the two. With respect to metabolic outcomes, predominant SCFA outputs were largely determined by resistant glucan class (mixed linkage α-glucans, resistant maltodextrins, and polydextroses), although some donor-dependent differences were observed among individual glucans within a class. Importantly, whether propiogenesis or butyrogenesis dominated in fermentation of a resistant glucan class was idiosyncratic to donors and inversely correlated, suggesting that (1) microbiomes have potential to produce either SCFA, but that (2) microbiomes are poised to produce one of the two from any given glucan class. It is important to account for differences in digestion of these glucans in the upper gastrointestinal tract, which may differentially alter the glucan structures finally encountered by gut microbiota [32]. However, these priming effects may underlie variable individual metabolic responses to dietary resistant glucans.

With respect to community composition, however, individual glucans within a class markedly diverged from one another in donor-dependent ways. Although we observed some consistency in responses across glucan classes at higher taxonomic levels, these relationships often shifted when individual glucans were considered independently. For example, members of genus *Bacteroides*, when considered at the level of glucan class, were overrepresented in cultures on resistant maltodextrins, with some species instead preferring polydextroses. When individual glucans were considered, however, this shifted to alternate preferences for resistant maltodextrins G and M, indicating that preferences of these OTUs for substrates within the resistant maltodextrin class was not equivalently high. Similarly, family *Lachnospiraceae* was a linear discriminant of polydextroses with respect to glucan classes, but of resistant maltodextrin D when all glucans were considered individually. Members of the genera *Bacteroides* and *Parabacteroides* as well as *Anaerostipes* (within *Lachnospiraceae*) have been observed to increase in abundance in feeding trials using some types of resistant starches [9,33,34,35,36,37,38]. In addition, these taxa appear to respond to resistant maltodextrins and polydextrose in some feeding trials (although often other fibers are co-administered, which may obscure the effect of the glucans) [18,32,39,40,41,42], so their general increases in abundance in cultures consuming resistant maltodextrins and polydextroses in our in vitro fermentations is not particularly surprising. However, to our knowledge, previous studies have not rigorously compared resistant glucans for differential effects on gut microbiome community structure. Intriguingly, however, variant resistant glucans have been observed to ferment different fecal SCFA concentrations in rat feeding trials [43], which suggests microbiome specificities as a hypothetical mechanism.

Underlying these higher-taxon differences, divergences among initial microbiota and glucan structures became even more pronounced when considered at the OTU level. To extend the previous example, relative abundance fold changes among *Bacteroides* OTUs varied as a function of both glucan and initial microbiota composition; although members of *Bacteroides* were responsive across individuals, the specific OTU-glucan associations varied. Interestingly, however, in many cases the same OTUs responded strongly to glucans across multiple donors (in some cases, all three) irrespective of initial abundances, suggesting that fundamental associations exist for some OTU-glucan pairs. Together, these data suggest that species may have preferences (possibly based on hydrolytic enzyme or transporter gene content) for particular glucan structures; however, it further suggests that an organism’s fitness on these substrates depends on competitive (or, possibly, cooperative) interactions with other species in the community. It should be noted that organisms can compete (and cooperate) in ways that are independent of carbohydrate consumption (for example, in exchange of terminal fermentation products [44,45]), and our study cannot separate these mechanisms of competition. However, to our knowledge, the degree of correspondence between specific OTUs (within responding higher taxa) for varying glucan structures and the influence of community context has heretofore not been demonstrated.

If our observations of strain specificity and context dependence in competition among fecal microbiota for variant resistant glucan structures in vitro are recapitulated in gut microbiomes in vivo, it suggests two related corollaries: (1) that outcomes of resistant glucan feeding trials are likely to initially be highly individual, based upon initial microbiome structure; but (2) over time, extended resistant glucan feeding should cause gut microbiomes to converge as the most efficient organisms increase in abundance across individuals. Further, it would suggest that this level of individual variability is likely to obscure outcomes of specific interactions between gut microbiota and variant fiber structures. Furthermore, as previous studies have asserted, it suggests that responses to well-defined fiber structures may be predictable given known microbiome community structures [46], and potentially even for novel fibers that have not yet been tested (by comparing their structures to microbiome responses to other structures). We submit that future experiments should endeavor to determine whether the specificities among different microbial species and glucan structures are recapitulated in feeding trials, and whether these differences result in divergent physiological outcomes.

## Figures and Tables

**Figure 1 nutrients-13-02924-f001:**
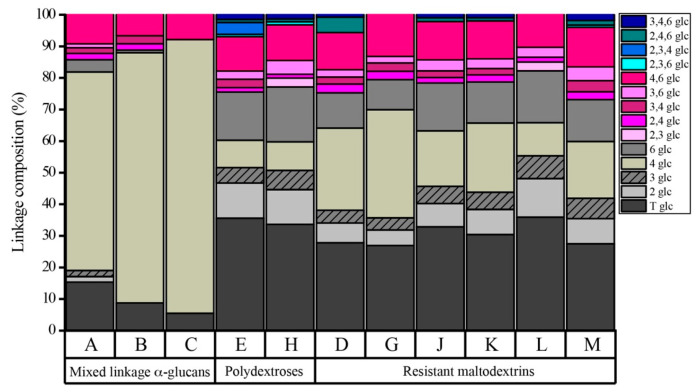
Glycosidic linkage composition of glucans, determined by the partially methylated alditol acetate method with gas chromatography-mass spectrometry. Shades of brown represent non-branched regions; shades of pink represent single branches and blue represents multiply-branched glycosyl residues.

**Figure 2 nutrients-13-02924-f002:**
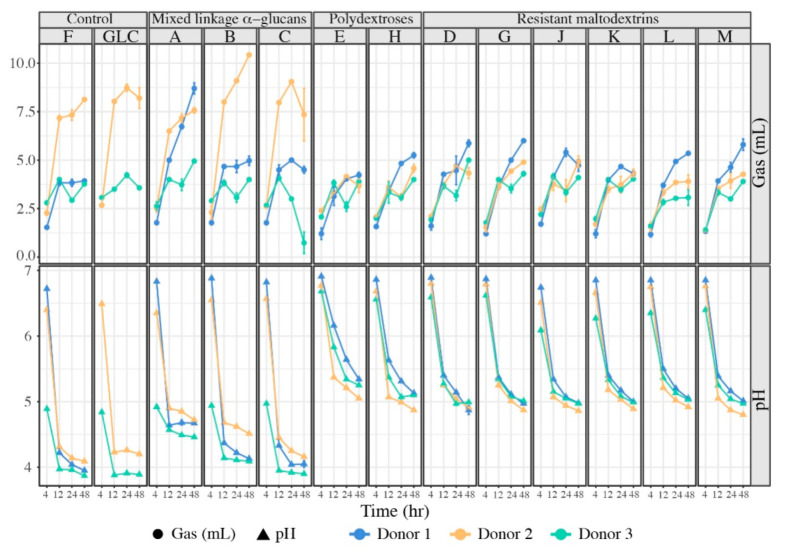
Gas (mL) and acid (pH) productions measured at 4–, 12–, 24–, and 48 h time points for all glucans across donor microbiota (blue: donor 1, orange: donor 2, green: donor 3). Individual glucans are represented by their letter designation; fructan (F) and glucose (GLC) were provided as controls. Error bars represent standard deviation of the mean of three separate replicates.

**Figure 3 nutrients-13-02924-f003:**
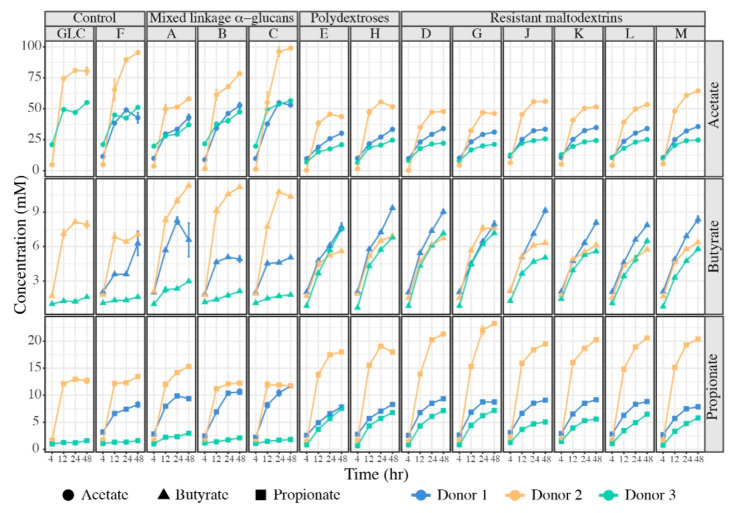
SCFA production at 4, 12, 24- and 48 h post-inoculation measured by gas chromatography. Different colors represent distinct responses of donors’ microbiota (blue: donor 1, orange: donor 2, green: donor 3) and different shapes represent the different SCFAs (acetate: circles, butyrate: triangles, propionate: squares). Individual glucans are represented by their letter designation; fructan (F) and glucose (GLC) were provided as controls. Error bars represent the standard deviation of the mean of three separate replicates.

**Figure 4 nutrients-13-02924-f004:**
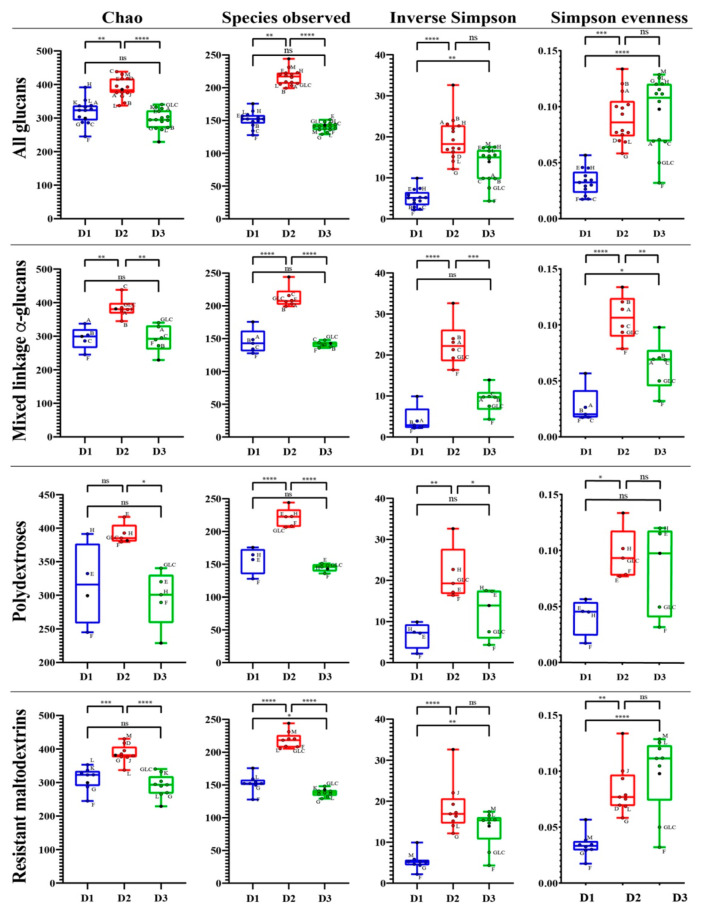
Comparison of alpha diversity metrics for community richness, evenness, and overall diversity among donors and within glucan categories, the initial fecal inoculum, and positive control groups containing glucose (GLC) and a fructan (F). Blue represents donor 1 (D1) microbiota, red represents donor 2 (D2) microbiota and green represents donor 3 (D3) microbiota. Measures of initial fecal samples are indicated by black filled dots; different glucans are indicated by their letters. For clarity, labels of glucans falling within the middle quartiles of the box and whisker plots are omitted from all glucan and resistant maltodextrin comparisons. Asterisks indicate statistical significance (* = *p* < 0.05, ** = *p* < 0.01, *** = *p* < 0.001, and **** = *p* < 0.0001; ns = not significant at *p* < 0.05).

**Figure 5 nutrients-13-02924-f005:**
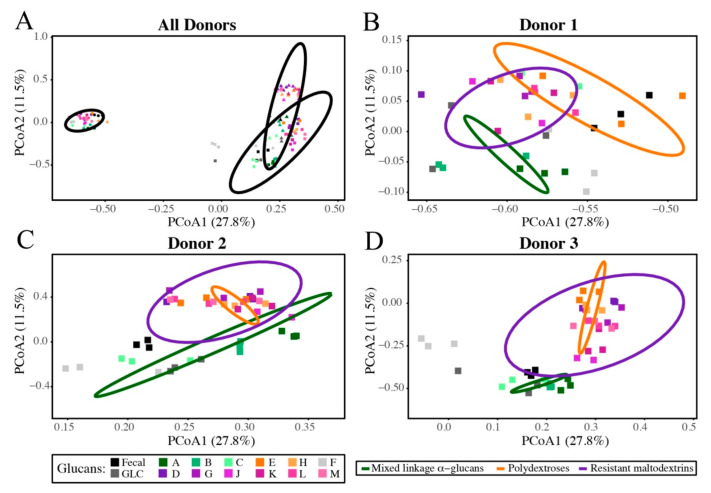
Principal coordinate analysis (PCoA) plot displaying differences in β-diversity, as calculated using the Yue and Clayton theta metric. Shapes represent different donors’ microbiota (all donors, (**A**); donor 1, (**B**); donor 2, (**C**); and donor 3, (**D**)) and colors represent the distinct glucans, within their glucan categories (mixed linkage α-glucans, greens; polydextroses, oranges; resistant maltodextrins, purples). Ellipses drawn based on 99% confidence level.

**Figure 6 nutrients-13-02924-f006:**
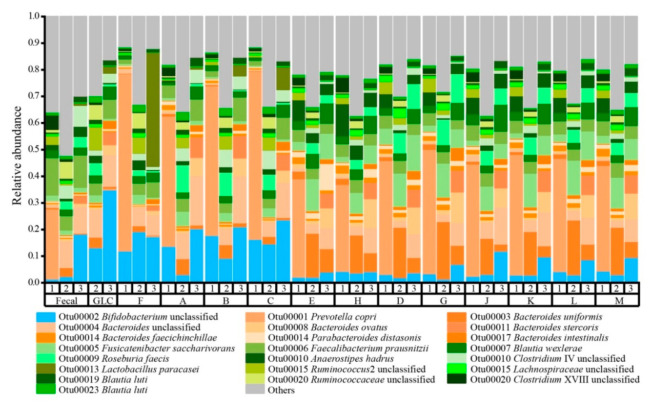
Relative abundances of dominant operational taxonomic units (OTUs ≥ 1%) after 48 h of fermentation of distinct glucan structures by donor microbiota and glucan. D1, D2, and D3 represent Donor 1, Donor 2, and Donor 3 inocula, respectively. Letters indicate different glucans, except for the glucose positive control (GLC) and the fructans positive control (F). Shades of green represent members of phylum *Firmicutes*, shades of orange represent members of phylum *Bacteroidetes*, and shades of blue represents members of phylum *Actinobacteria*. Taxa observed at a relative abundance of <1% relative abundance are combined into “Other”.

**Figure 7 nutrients-13-02924-f007:**
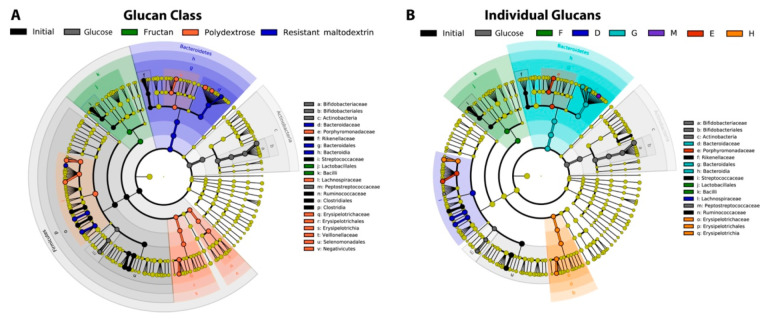
Linear discriminant analysis of bacterial taxa differentiating glucan classes (**A**) and glucans (**B**) (both LDA ≥ 3.5). Nodes represent different taxonomic levels increasing in taxonomic resolution (species on the outer ring). Resistant maltodextrins are depicted as shades of blue, and polydextroses as shades of orange. No significant associations of taxa with mixed linkage α-glucans were observed at this LDA threshold.

**Figure 8 nutrients-13-02924-f008:**
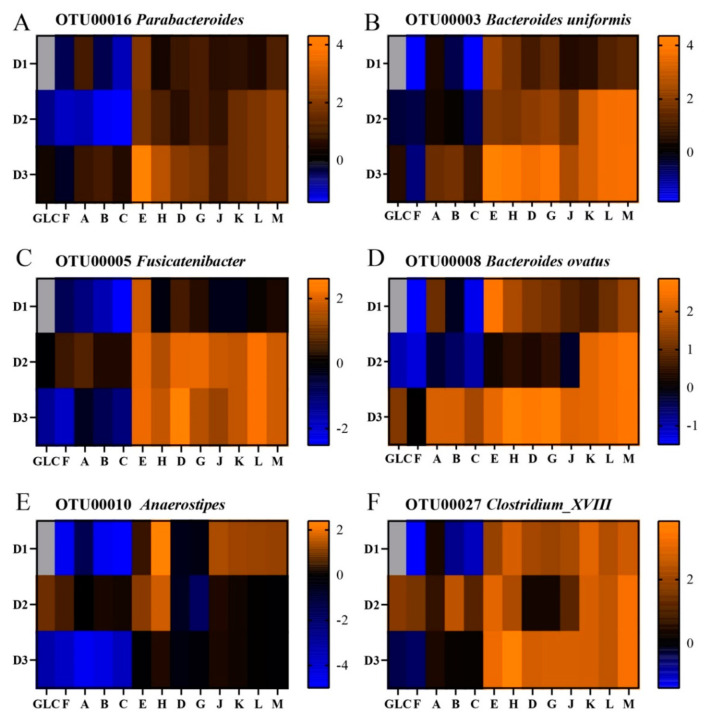
Log_2_-transformed fold change of OTU abundances after glucan fermentation (48 h post-inoculation relative abundances divided by fecal relative abundances). The fold changes of the six most responsive OTUs (OTU00016, (**A**); OTU00003, (**B**); OTU00005, (**C**); OTU00008, (**D**); OTU00010, (**E**); and OTU00027, (**F**)) are depicted across all donors. Orange shades represent increases in abundance; blue shades represent decreases in abundance; black indicates no change with respect to the initial microbiota. OTUs not detected within an individual’s inoculum are gray.

**Figure 9 nutrients-13-02924-f009:**
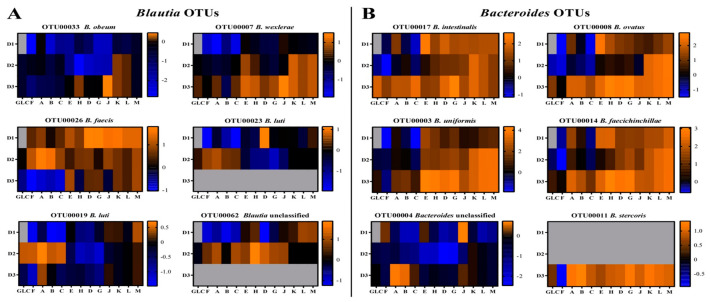
Log_2_-transformed fold change of OTU relative abundances within genera *Blautia* (**A**) and *Bacteroides* (**B**) after glucan fermentation across three donor communities. Orange shades represent increases in abundance; blue shades represent decreases in abundance; black indicates no change in abundance with respect to the initial microbiota. OTUs not detected within an individual’s inoculum are gray.

**Table 1 nutrients-13-02924-t001:** Analysis of Molecular Variance (AMOVA) showing separation of glucan categories for each donor (D1, D2, D3), calculated by mothur with a default alpha of 0.05 and Bonferroni correction for multiple comparisons. * Indicates statistically significant differences between the glucan categories.

	Initial			Control			Mixed Linkage α-Glucan			Resistant Maltodextrin		
	D1	D2	D3	D1	D2	D3	D1	D2	D3	D1	D2	D3
Initial	-	-	-	0.024	<0.001 *	0.036	0.003	<0.001 *	<0.001 *	<0.001 *	<0.001 *	<0.001 *
Control	0.024	<0.001 *	0.036	-	-	-	0.014	<0.001 *	0.012	<0.001 *	<0.001 *	<0.001 *
Mixed linkage α-glucan	0.003	<0.001 *	<0.001 *	0.014	<0.001 *	0.012	-	-	-	<0.001 *	<0.001 *	<0.001 *
Resistant maltodextrin	<0.001 *	<0.001 *	<0.001 *	<0.001 *	<0.001 *	<0.001 *	<0.001 *	<0.001 *	<0.001 *	-	-	-
Polydextrose	0.019	0.008	0.004	0.005	<0.001 *	<0.001 *	<0.001 *	<0.001 *	<0.001 *	<0.001 *	0.232	<0.001 *

## Data Availability

Sequence data are available in the National Center for Biotechnology Information’s Sequence Read Archive under BioProject PRJNA721693 as BioSamples SAMN18744139–SAMN18744255.

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
