# Peer review of "Fine Carbohydrate Structure of Dietary Resistant Glucans Governs the Structure and Function of Human Gut Microbiota"

_nutrients, 2021, doi:10.3390/nu13092924_

Round 1

Reviewer 1 Report

In general:

In a paper by Romero Marcia1 et al. it was stated that only minor differences in the binding structure of glucans could affect the composition of the gut microbiota and fermentation products. To this reviewer, the authors' original title seems to be an expansive interpretation of the study results and a forced universalization. The title should be changed to directly reflect the results of the study, or the authors should be asked to rebut it.

In specific:

(1) As the authors stated in the abstract, the final goal of this study is to support the rational inclusion of specific fibers in dietary patterns to modulate the gut microbiome in support of health. Normally, major fermentation substrates entering the colon are natural plant fibers (dietary fibers in the narrow sense of the definition), resistant starch, and the endogenously secreted mucins. Resistant dextrin derivatives (polydextrose and resistant maltodextrins) are artificially synthesized fibers. Based on these evaluations alone, would it make sense to induce the ideal gut microbiota in terms of public nutrition?

(2) As for in vitro fecal fermentation, isn't it necessary to adjust pH during fermentation? According to the findings of Cummings et al., pH values in the cecum and proximal colon are not less than 5.0. An extreme reduction of pH may induce important consequences for the composition of microbiota and the balance of microbial metabolites (Appl Environ Microbiol 2005; 71:3692-700. Environ Microbiol 2009; 11:2112-22). 

Reviewer 2 Report

Very interesting paper and I like it very much. Some comments:

Line 25 (Abstract): "operational taxonomic units (OTU)"

Lines 80, 249, 268, 271, 274 (and other similar lines): mixed linkage α-glucans

Lines 113-116: The GC-MS protocol you wrote is different from Pettolino. Was this method published before ?

Methods section in general: If the Methods have been published before, please cite previous literature.

Lines 117-125: How did you choose the fecal donor ? What were your criteria ?

Line 162: 100 ng/μL

Lines 183-195: Please cite Kozich et al (2013) if you used mothur.

Figure 2: What does GLC mean ? Please check the Figure caption as well.

Figure 2: Why is there no donor 1 in "GLC" Control ?

Lines 210-211: Why did you use this ordering of letter labels ? Is there a rationale behind this ? 

Lines 253-254: "The decrease in pH was especially marked in fermentation of glucans A-C by donor 3 microbiota." - Should it be donors 1 and 2 ? There was a more rapid change in pH in donors 1 and 2.

 A comment out of curiosity on lines 321-332: Interesting what OTUs are present based on the diet. I would assume, since donor 2 has more plant-based diet, that the microbiota from donor 2 will prefer the fermentation of the glucans, in general. This is probably reflected by the results in Figure 3 where more SCFAs were produced by donor 2 ?

A comment out of curiosity on lines 436-438: Can you culture the bacteria separately and perform the digestion of these glucans to remove the effect of the interaction with other community members ?

A comment out of curiosity on lines 389-395: Since glucan A-C quite resemble starch, does it mean that there could be no microbe discriminant for starch ? (I hope my question makes sense...) Or, to rephrase it, starch is digested/fermented differently from other glucans if it survives at all and reaches the gut ?

Line 482: "to variables" to "two variables"

General comment: A good paper that warrants further and more in-depth studies because it raises a few questions, such as: how big the effect of coexisting bacteria (also strain type) on the degradation of the glycosidic linkage ? Can you isolate bacterial types and see how glycosidic linkages are degraded ? 

Additional comments:

  1. Would your results apply to dietary fibres where various polysaccharides are present in a complex mixture ?
  2. What happens if you mix all the glucans you used in your experiments and ferment the mixture using the fecal matter ? Would you still be able to observe discrimination amongst the different fibres and microbial species ?
